# The Role of “Critical” Ultrasound Reassessment in the Decision-Making of Bethesda III Thyroid Nodules

**DOI:** 10.3390/medicina59081484

**Published:** 2023-08-17

**Authors:** Giuseppina Orlando, Giuseppa Graceffa, Sergio Mazzola, Fabrizio Vassallo, Maria Pia Proclamà, Pierina Richiusa, Stefano Radellini, Nunzia Cinzia Paladino, Giuseppina Melfa, Gregorio Scerrino

**Affiliations:** 1Unit of General and Emergency Surgery, Department of Surgical Oncological and Oral Sciences, Policlinico “P. Giaccone”, University of Palermo, Via Liborio Giuffré 5, 90127 Palermo, Italy; giusi_orlando@hotmail.it (G.O.); fabris.fv@gmail.com (F.V.); mariapiaproclama@gmail.com (M.P.P.); irene_melfa@yahoo.it (G.M.); 2Unit of Oncological Surgery, Department of Surgical Oncology and Oral Sciences, University of Palermo, Via L. Giuffré, 5, 90127 Palermo, Italy; giuseppa.graceffa@unipa.it; 3Unit of Clinical Epidemiology and Tumor Registry, Department of Laboratory Diagnostics, Policlinico “P. Giaccone”, University of Palermo, Via L. Giuffré, 5, 90127 Palermo, Italy; mazzolasergio3@gmail.com; 4Department of Health Promotion Sciences Maternal and Infantile Care, Internal Medicine and Medical Specialties (PROMISE), Section of Endocrinology, University of Palermo, 90127 Palermo, Italy; pierina.richiusa@policlinico.pa.it (P.R.); stefano.radellini@policlinico.pa.it (S.R.); 5Department of General Endocrine and Metabolic Surgery, Conception Hospital, Aix-Marseille University, 147, Boulevard Baille, 13005 Marseille, France; 6Unit of Endocrine Surgery, Department of Surgical Oncological and Oral Sciences, University of Palermo, Via L. Giuffré, 5, 90127 Palermo, Italy; gregorio.scerrino@policlinico.pa.it

**Keywords:** Bethesda III, ultrasound, thyroid malignancy, cytology

## Abstract

*Background and Objectives*: Bethesda III (BIII) thyroid nodules have an expected malignancy rate of 5–15%. Our purpose was to assess which US criteria are most associated with cancer risk, and the value of critical ultrasound (US) reassessment. *Methods*: From 2018 to 2022, 298 BIII nodules were enrolled for thyroidectomy (79 malignancies). We evaluated ultrasonographic data: hechogenicity, intralesional vascularization, spiculated margins, micro-calcifications, “taller than wide” shape, extra-thyroidal growth, size increase, as well as their association with histology. We also evaluated if the ultrasound reassessment modified the strategy. *Results*: Spiculated margins and microcalcification were significantly correlated with malignancy risk. Spiculated margins showed a specificity of 0.95 IC95% (0.93–0.98); sensitivity 0.70 IC95% (0.59–0.80). Microcalcifications showed a sensitivity of 0.87 CI95% (0.80–0.94); specificity 0.75 CI95% (0.72–0.83). The presence of these signs readdressed the strategy in 76/79 cases Then, the indication for surgery was appropriate in 75% of cases. *Conclusions*: Microcalcifications and spiculated margins should be routinely sought during a final ultrasound reassessment in BIII nodules. These signs allowed for a modification of the strategy in favor of surgery in 96% of the cases that were not otherwise referred to surgery. The importance of integrating ultrasound and cytology in the evaluation of BIII thyroid nodules is confirmed. Reassessment with ultrasound of BIII nodules allowed for a redirection of the surgical choice.

## 1. Introduction

Thyroid nodule is a common clinical problem whose evaluation is commonly performed by ultrasound (US) and cytology [1]. The widespread use of US has raised the prevalence of thyroid nodules to 50% of the general population. Malignancy is found in just over 9% of solitary nodules and 6.3% of multinodular goiters [2]. It is influenced by several risk factors [3]. Depending on the epidemiological procedure applied, probability that a nodule conceals a carcinoma, albeit mostly indolent-behaving as the papillary variant, can be as high as 15% [4,5].

A reporting system for the cytopathological diagnosis of thyroid nodules (Bethesda System) was published in 2009 [6] and is still in use as of its 2017 update [7]. This system identified six diagnostic categories, among which, Category III (globally identified as ‘indeterminate low-risk’) constitutes a heterogeneous group of alterations containing nuclear/cellular or architectural atypias, otherwise mentioned respectively as “atypia of uncertain significance” (AUS) and “follicular lesions of uncertain significance” (FLUS). In fact, it seems to be clear that AUS would present a higher risk of carcinoma than FLUS [8]. However, some studies emphasized that Bethesda Category III also contains, among other scenarios, possible sampling artifacts, and caution is suggested in suspecting a specimen in which nuclear atypia coexist with autoimmune thyroiditis, or in the presence of some isolated Hürthle cells in the context of other normal cells, or in the presence of a conspicuous proportion of colloid [9]. An extensive literature review by Kholová and Ludvìková (2014), in emphasizing the heterogeneity of the BIII category, claims its clear cytomorphologic identity and advocates improved diagnostic potential with molecular analysis, immunohistochemistry, or even imaging [10].

As a result of its heterogeneity, the BIII category is one of the major concerns in thyroid cytology, despite the fact that this diagnostic method is to be considered the most accurate investigation in thyroid nodule evaluation [11]. The approach to this diagnostic class therefore is managed in different ways: observation and follow-up, repeat investigation with possible molecular tests, ‘precautionary’ surgery. These choices are guided both by the team’s experience and by the patient’s preferences on the basis of informed consent [12].

Several studies have highlighted the role of US in providing useful information for a more accurate reassessment of cancer suspicion in focal lesions for which cytology has given an inconclusive result [12,13].

The main US features increasing oncologic risk of thyroid nodules of the most applied US-based systems are microcalcifications, irregular margins, taller-than-wide shape and marked hypoechogenicity. These features are all incorporated in the most widespread US risk score system in Europe, the five-steps EU-TIRADS. This system demonstrated increasing PPV from the early stages (near 0) to stage 5 (PPV > 30%) [14].

It has long been known that cytology is able to discriminate a substantial number of patients with a high probability of malignancy from subjects who have a very low probability (3%) of falling within it [15]. However, the indeterminate results suggest the search for solutions to overcome this problem.

In this study, we wanted to examine the data available at our institution. The data collected over a five-year period from outpatient recruiting subjects suffering from thyroid nodule were thus examined. The primary endpoint was to highlight which ultrasound signs are most associated with the diagnosis of carcinoma in the low-risk indeterminate thyroid nodule (BIII). Another purpose of this study was to evaluate whether and how much a “critical” reevaluation of ultrasound, repeated after obtaining the result of the cytological examination, would change the choices in favor of surgery in BIII nodules and in particular which of the specific ultrasound signs are mainly involved in the decision change.

## 2. Materials and Methods

This is a retrospective study carried out at our institution (Department of Surgical, Oncological and Stomatologic Disciplines of Policlinico “P. Giaccone” in Palermo) from 1 January 2018 to 31 December 2022. In this period, among 1292 thyroidectomies performed, 369 papillary thyroid carcinomas (PTC) were found. Of these, 79 were classified as the cytological diagnostic class “Bethesda III” (BIII), 184 as “Bethesda IV” (BIV), 67 were classified as “Bethesda V” (BV), and 21 as “Bethesda VI” (BVI). Finally, 18 constituted incidental findings (Nodules classified Bethesda I or II, or occasional findings in the context of multinodular goiters or Graves’ disease that did not undergo FNAB). During the same reporting period, a total of 298 nodules classified as Bethesda III underwent surgery at the same institution. This group included both the 79 PTCs mentioned above, as well as other diagnoses. Therefore, all patients with a Bethesda III cytology subsequently subjected to surgery were recruited for this study. The flowchart in Figure 1 shows how patients were recruited for the study. The 79 operated PTCs from the BIII diagnostic category are part of the total 369 operated PTCs, just as the total 298 BIIIs are part of the 1292 thyroidectomies performed during the reporting period.

At the time of ultrasonographic reevaluation, the analysis was based on the ultrasonographic features, namely following the indeterminate results.

Patients with endocrine disorders are usually treated by an institutional multidisciplinary team. The institutional multidisciplinary team for endocrine diseases is made up of professionals such as endocrinologists, endocrine surgeons, specialists in nuclear medicine, pathologists, otolaryngologists, specialists in medical oncology, and nephrologists. In particular, those enrolled in the present study were operated on by high-volume surgeons, according to the commonly accepted criteria of the scientific reference societies [16,17]. Patients were enrolled for surgery after evaluation by this team.

All patients with a BIII cytology subsequently subjected to surgery (thyroid lobectomy or total thyroidectomy) were included in this study.

### 2.1. Clinical Management

The ultrasound investigation was carried out in the context of the team by an experienced endocrinologist (PR or SR) or endocrine surgeon (GS, GM, GO or GG).

Patients with two (or, exceptionally, more) nodules with BIII-IV-V-VI results were excluded from the study because they were considered a potential confounding factor and the analysis would be more complicated in such cases. We would have excluded US reports not in agreement with the AACE/ACE/AME standards, but in any case, the descriptions carried out at our institution were always performed strictly following these criteria. The cytopathologic reports were always complete and in accordance with the Bethesda system. Moreover, we excluded benign thyroid diseases (Grave’s disease) and carcinomas other than papillary (follicular, Hürthle-cell, medullary thyroid carcinomas, and all malignancies other than those differentiated of follicular origin) in order to make the analyzed case series homogeneous and to exclude potential bias. With the same intention, highly aggressive forms of PTC were also excluded. We also excluded patients with personal histories of neck irradiation.

The FNAB result was related to the same nodule that had been biopsied, without considering the overall result of the whole specimen.

In accordance with current guidelines [11], only nodules > 1 cm in size underwent cytological examination. Therefore, the finding of a microcarcinoma in the specimen was considered occasional, which was well individualized from the nodule undergoing FNA. In any case, only nodules > 1 cm were included and evaluated in the present study.

None of the nodules included in this analysis were classified as non-invasive follicular thyroid neoplasm with papillary-like nuclear features (NIFTP), thyroid tumors of uncertain malignant potential, or hyalinizing trabecular tumor.

In summary, the inclusion/exclusion criteria were as follows:

Inclusion:-any ‘non aggressive’ PTC recruited in BIII diagnostic category (2018–2022);-US performed by endocrinologist/endocrine surgeon of multidisciplinary team.

Exclusion:-two or more BIII-IV-V-VI nodules;-US reports not in agreement with ACE/AACE/AME standards;-cytology reports not in agreement with Bethesda system;-Grave’s disease and other benign diseases;-cancers other than PTC;-aggressive variants of PTC;-NIFTP and other tumors of uncertain malignant potential;-PTC ≤ 1 cm.

### 2.2. Thyroid Ultrasound

Ultrasound machines equipped with a high-frequency probe (7.5–12 mHz) were used. The ACE/AACE/AME score system, already in use at our institution since 2016, was applied for risk assessment [18]. Ultrasonographic investigation was used to target nodular lesions for cytologic examination when the exam was originally performed at our institution (161 patients = 54%). In this case, the indication for cytological examination was given by the finding of a lesion with intermediate/high risk score. The remaining 137 patients (46%) were from other institutions or from endocrinologists in the territory, and in case of the latter, we only took note of the cytological examination exhibited. Instead, US assessment was repeated by the multidisciplinary team.

The first US assessment was performed, therefore, at the time of patient recruitment by the multidisciplinary team. US reassessment was performed after obtaining the results of cytology. The US features reported in the analysis were captured at the time of reassessment.

### 2.3. Thyroid Cytology

Fine-needle aspiration biopsy (FNAB) was always performed under ultrasound guidance using a 25-gauge needle. Cell sample collection was always by capillary and never by aspiration. Once the collection was complete, the fluid was transferred to slide initially by surface tension, then gently pushed with the syringe plunger. For each nodule, 4 to 8 slides were packed, immediately fixed with isopropanol on polyethylene glycol support, set in spray formulation, and sent immediately to the cytology diagnostic service of our institution.

### 2.4. Thyroid Surgery

Each BIII nodule taken over was treated with complete lobectomy extended to the thyroid isthmus. The surgery was always performed according to current dictates of proper technique, with selective ligation/sectioning of thyroid artery branches. An energy device was almost always employed. In the presence of concomitant nodules affecting the contralateral lobe, a total thyroidectomy was performed. All procedures were assisted with nerve monitoring.

### 2.5. Variables Analyzed

We analyzed both demographic variables (age, sex) and the different aspects usually assessed at conventional ultrasonography, which are: nodule size, its nature (cystic/solid), echogenicity (iso/hyperechogenic, slightly hypoechogenic, frankly hypoechogenic), intralesional vascularization (present/absent), spiculated margins (present/absent), intranodular microcalcifications (present/absent), “taller than wide” shape (present/absent), extrathyroidal growth (present/absent), and volume increase in two or more evaluations over time (present/absent). Thus, all variables were dichotomous except age (continuous) and echogenicity, which, having three possible outcomes, was recorded as multifactorial.

### 2.6. Statistical Analysis

In a first step, a univariate analysis was carried out. Fisher’s exact test for categorical variables and Welch’s parametric test for continuous variables were applied.

The variables age and gender together with the variables found to be statistically significant in the univariate analysis were entered into a logistic multivariate model.

Subsequently, the stepwise method made it possible to choose the best model (significance of the regression coefficients and the lowest AIC [Akaike Information Criterion] value). The model adopted (otherwise referred to as stepwise analysis) allows for the isolation of variables that are actually independent of each other, while excluding those with possible intercorrelation. Thus, the variables possibly excluded are not only those that are not significant to multivariate analysis, but also those that may be at risk of interdependence.

Subsequently, sensitivity and specificity were calculated on the variables that were found to be statistically significant according to this second analysis.

Statistical calculations were carried out with the software RStudio (version 3.4.1 of 30 June 2017) for R (version 2.1) [19,20]. Specificity and sensitivity were calculated using the “pROC” application package.

The present study was performed as per the declaration of Helsinki (1964) and its amendments. Informed consent was obtained from all patients.

## 3. Results

Among the 298 nodules enrolled in the study, 219 were benign, while papillary carcinoma was found in 79. The malignancy rate among the BIII nodules in the evaluated patient series was, therefore, 26.5%.

In Table 1, we report the univariate analysis for each variable analyzed to quantify its association with outcome (benign/malignant nodule). At this first step, all variables analyzed, excluding age and sex, showed a statistically significant association with malignancy.

Table 2 shows the results of the “critical” re-evaluation, carried out on 287 patients, and the subsequent change of strategy.

Table 3 summarizes the data from the multivariate analysis, relating to the variables found to be most significant at the Akaike Information Criterion (AIC). This second analysis did not confirm this association for all variables.

In fact, among the ultrasonographic signs analyzed, we found two that perform best in indicating the risk of malignancy: irregular margins and microcalcifications.

In particular, irregular margins appear to be a more specific than sensitive indicator, with the following values, respectively: Specificity 0.95 CI95% (0.93–0.98); Sensitivity 0.70 CI95% (0.59–0.80). Microcalcifications are a more sensitive than specific indicator, with the following values, respectively: Sensitivity 0.87 IC95% (0.80–0.94); Specificity 0.75 IC95% (0.72–0.83) (Figure 2).

Figure 2 and Figure 3 represent ROC curves for the two variables most correlated with malignancy.

Figure 4 shows the box plot regarding the size distribution of benign nodules undergoing ultrasound for “critical review” purposes. The graph shows that the median size of the distributions of the 186 benign nodules is 4 cm; while the first quartile turns out to be 3.5 cm; therefore, the possible cut off below which surgery should not be performed could be identified between the values 3.5 cm–4 cm, which is in agreement with the literature data and the main available guidelines [21]. In agreement with our data, surgery performed according to the correct guidelines falls in the range (219–137 = 82 + 79 malignant) = 161; (161/298) = 54% and (219–85 = 134 + 79 malignant) = 213 (213/298) = 71.5%).

Taking malignancy into consideration, “critical reassessment” directed a change of strategy in 76/79 cases, i.e., 96% of cases, while in only 22/219 cases, benign nodules were considered an appropriate indicator that the choice should be changed after such reassessment.

Small size might appear to be a highly significant risk factor, but this finding is marred by selection bias, as large nodules (generally, near or greater than 4 cm in diameter) are taken into consideration for thyroidectomy, regardless of malignancy. Therefore, it is consequently obvious that the category of benign nodules consists of lesions of larger volume on average than malignant ones.

## 4. Discussion

The BIII cytology diagnostic category still constitutes a grey area. A number of alterations coexist in this category [16]. Its expected malignancy rate ranges from 5 to 15% [22]. Nine specific groups of changes correspond to AUS/FLUS alterations: microfollicular organization, variable representation of Hurtle cells, cellular atypia attributable to preparation artefacts, and focal cellular atypia that may take on papillary carcinoma-like aspects [9]. Mosca and coll. (2018) showed that the risk of malignancy of the AUS category was not only higher than FLUS (27% versus 5.6%), it was also higher than the Bethesda IV category, whose risk in their sample was 16% [23].

In a study by Kaymaz et al., (2020) [24], atypia was further divided into nuclear, architectural, Hürthle-cell change, and others, and the nuclear ones were analyzed in even more detail. The final result was in agreement with previous literature data (nuclear atypia was the most associated with malignancy), but for some specific nuclear atypia, the association was as high as 100% [24]. However, in real scenarios, such details are difficult to obtain because of the variability of cytologists’ reports in different settings, or because samples with heterogeneous cellularity do not allow for reliable standardization.

Especially in the last two decades, cytology has been enriched with molecular biology tests, which have improved its initial diagnostic performance [19,25,26].

Among the genetic mutations that can be assessed in thyroid carcinoma, the somatic B-RAFV600E mutation is the best known [20]. This mutation is present in several human cancers, including papillary thyroid carcinoma, and appears to be associated with a poor prognosis, albeit with relatively low specificity [27,28]. Further mutations that can be found in FNAB samples are those related to the telomerase reverse transcriptase (TERT) promoter [29]. Mutations of rat sarcoma viral oncogene homolog (RAS) and rearrangement of paired box gene 8-peroxisome/proliferator-activator receptor γ (Pax8/PPARγ) have also been tested to improve the performance of cytology [30]. TERT mutation appears to be associated with more aggressive forms of the tumor. RAS mutation, as well as some others, has even more value, especially if multiple mutations are found in association [30,31].

In clinical practice, molecular biology tests have some limitations. In fact, they all have a high throughput if the test is positive, but their negativity is poorly predictive. Moreover, they are generally rather expensive tests, and on the other hand, it is not clear whether and to what extent their systematic application can reliably reduce the incidence of unnecessary surgery [32].

Ultrasonography has traditionally been used as a first-level investigation (in practice, immediately after or even in association with clinical examination). In current clinical practice, the most common risk scores have been used for the purpose of referring detected lesions to FNAB or follow-up [8,33,34].

Several recent studies have shown that thyroid ultrasonography can also improve the predictive value of lesions found to be indeterminate at cytology [35,36,37,38].

A study carried out at our institution found a close correlation between malignancy and ultrasound risk score, assessed according to ACE/AACE/AME criteria [38], in the context of nodules with indeterminate cytology (BIII and IV). Although this correlation was generally 1.5 times stronger in BIVs than in BIIIs, the risk of malignancy was nevertheless 10 times higher in the ultrasound classes of highest ultrasound risk (intermediate and high) than in the ultrasound class of low risk, even in the context of BIII nodules. This correlation was more reliable if the ultrasound examination was performed by a member of the multidisciplinary team (endocrinologist, endocrine surgeon) rather than by a generalist sonographer [39].

The use of a more widespread score system such as K-TIRADS leads to very similar results, suggesting, in clinical practice, a more accurate selection of patients to be referred for surgery in Bethesda III nodules with low-risk ultrasound scores [8,40].

The recent systematic review with meta-analysis by Gao et al. found a pooled sensitivity of 0.75 (95% CI 0.72–0.78) and a pooled specificity of 0.48 (95% CI 0.45–0.50) for ultrasonography in evaluating the risk of malignancy of BIII Nodules. With two or three signs of malignancy, the sensitivity and specificity were 0.77 (95% CI 0.71–0.83) and 0.54 (95% CI 0.51–0.58), 0.66 (95% CI 0.59–0.73), and 0.71 (95% CI 0.68–0.74), respectively. In this study, ultrasound improved the ability to differentiate benign from malignant BIII nodules. The greater the number of ultrasound signs of suspicion, the greater the ability of ultrasound to target malignancy [12].

Other studies have assessed the risk of BIII nodules, not only by conventional US criteria, but also stiffness assessed by elastosonography [41].

Multiparametric US criteria were also evaluated in another study, in which two-dimensional B-mode US (2B), qualitative strain elastography, and a planar and volumetric (3D) color Doppler using both a linear multifrequency probe and a linear volumetric probe were used. The identification of highly suspicious US characteristics (marked hypoechogenicity, irregular margins, intranodular inhomogeneity, taller-than-wide shape, intranodular microcalcifications, lymph nodes with hilum changes, moderate and severe stiffness showed at elastosonography, moderate or marked perinodular vascularity) observed with all techniques increased the risk of malignancy [42].

More recently, it has been shown that the management of Bethesda category III nodules can be optimized by combining repeat cytology with the ultrasound score system recommended by the European thyroid association (EU-TIRADS). It is also suggested in this study that FLUS lesions and AUS should be assessed separately [43]. Nevertheless, the higher risk of malignancy of nodules with cellular atypia does not reach statistical significance [23,44,45].

On the other hand, the extreme variability in the incidence of carcinoma on BIII lesions is well demonstrated. The reasons for this variability are correlative to selection bias that make the patients recruited in the endocrinological setting very different from the surgical ones, and these differences also depend on the criterion of patient selection for cytology (clinical, ultrasound in its different aspects), but also for the subsequent surgical choice (clinical, ultrasound, criteria for distinction between AUS and FLUS, search for mutations, etc.) [44,45,46].

In our study, the ultrasonographic signs that seem most associated with “critical” reassessment at a second ultrasonography, and most associated with carcinoma risk, were spiculated or microlobulated margins and microcalcifications. These findings correlate well with those of other studies, in which microcalcifications, nodule shape, margin irregularity, and hypoechogenicity were found to be variously correlated with increased risk of malignancy [12,27,47,48].

In our study, we also included the volume increase over a time period among the variabilities. This can be defined as suggested in recommendation 23 of the 2015 ATA guidelines (20% increase in at least two nodule dimensions with a minimal increase of 2 mm or more than a 50% change in volume) [21]. Given the association between volume increase and RAS mutation in benign nodules, it may be useful to perform such genetic evaluation in the future [49].

This variable, found to be significant at univariate analysis, was not taken into account according to the AIC assessment; however, it remains an important parameter at the time of indication for performing FNAB.

We believe that the two signs most significantly associated with the risk of malignancy have this behavior because of its focality. Since they affect a minority segment or area of the nodule, they may be observed at the “critical” reevaluation rather than in the immediacy of an investigation initially performed for screening purposes to target FNAB.

Concerning benign nodules, we observed that the average size of the excised nodules corresponds to 4 cm. (Figure 3). This size is indicated by the most widely used guidelines as the threshold value for surgical indication in benign nodules [21]. However, this indication should not be considered mandatory, first because of the low level of evidence for this recommendation, and second because clinical symptoms, disfigurements, or other conditions may arise even with smaller size, prompting patients to prefer surgery. The first quartile (3.5 cm.) appeared as a reasonable limit to indicate a threshold above which these symptoms may be frequent. Above this limit, patients (corresponding to 75% of benign nodules) had a most likely appropriate indication for surgery. In conclusion, in our series, surgery was an overtreatment in only 25% of cases. This probably occurred because of an overestimation of the symptoms reported by patients, or because of anxiety resulting from knowledge of a pathology that was not well characterized.

This study undoubtedly has some limitations. Apart from the retrospective model, it does not consider molecular biology investigations and does not evaluate nodules according to subclasses (AUS/FLUS) of BIII. Regarding molecular biology investigations, we believe that current costs do not allow for their widespread use. A further limitation is the low negative predictive value. Concerning the BIII subcategories, there are still uncertainties about the different levels of significance of individual atypia. This makes any assessment unreliable. On the other hand, the reclassification of class BIII is currently not an acquired standard, but rather a dutiful indication. Such a distinction is not usual in our routine, given the current uncertainties in this regard. It is, however, our intention to reevaluate, in the future, the results of our experience by applying the distinction between the subcategories of BIII, if this distinction becomes more reliable.

Another limitation of the study may be a selection bias resulting from the fact that the nodules taken into consideration were, in principle, “suspicious” for malignancy. This may not reflect the reality of all BIII nodules.

It can also be assumed that the first ultrasound examination, coming from different practitioners with uneven experience, and being performed in only a minority of cases within the institutional expert team, may employ different criteria for patient selection and the description of different signs. This may influence the subsequent patient’s clinical course, but it is certain that the “critical” re-evaluation, performed after obtaining the cytology result, was an important tool for setting evaluation criteria and achieving the same evaluation standard for all cases enrolled in the study, since the sonographer were dedicated and experienced, coming from a high-volume team.

Given the heterogeneous origin of the USs at the first evaluation, we did not compare individual variables before and after reassessment. Moreover, for some of them (e.g., “taller than wide” appearance, “extra-thyroid growth”), the absence of items in the benign category would have invalidated the results.

Finally, it should be emphasized that the study is limited to correlating ultrasound and cytology in the context of ‘indolent’ variants of PTC (classical and follicular, encapsulated or not), whereas variants with more aggressive potential, such as diffuse sclerosing, tall cells, and others, were not evaluated, given the bias that could arise from their rarity. For the same reason, since histotypes other than papillary were excluded, little can be said in this regard.

Our study does not intend to conclude with a debasement of the value of ultrasound signs whose association with increased cancer risk is well established. The statistical analysis performed identified and evaluated the nondependent variables. We do not exclude that further studies with a larger number of enrolled patients may identify additional significant variables.

As a further result, “critical” reassessment should be noted. It has proven to be effective in 96% of nodules found to be malignant at definitive histology.

Ultimately, these results come from a real-world scenario, and may provide a starting point for further studies that include other criteria of evaluation.

## 5. Conclusions

In conclusion, this study once again confirms the importance of integrating ultrasound and cytological examination in the evaluation of so-called indeterminate thyroid nodules. This pathway is capable, at least on the theoretical level, of considerably reducing unnecessary thyroidectomies with a sufficient degree of safety.

It is also true that a more careful interpretation of indeterminate cytology reports, selecting the most suspicious cellular atypia, could lead to a similar improvement in the management of BIII thyroid nodule; but to date, it is not fully clarified how directly transferable this is to clinical practice.

As shown by our results, the change of strategy resulting from the reassessment of USs after obtaining cytology can reveal underestimated PTCs at the first ultrasound examination. On the other hand, this means that even small details of the US examination can change the surgical choice when faced with an “indeterminate” finding.

In this light, we believe that at present, the integration of imaging and cytology is the gold standard for the correct interpretation of thyroid nodules. Future studies could quantify the number of thyroidectomies avoided, complications prevented, costs reduced, and quality of life improved in correlation with this integration. However, improved interpretation of thyroid cytology and greater accessibility of associated molecular biology remain desirable.

## Figures and Tables

**Figure 1 medicina-59-01484-f001:**
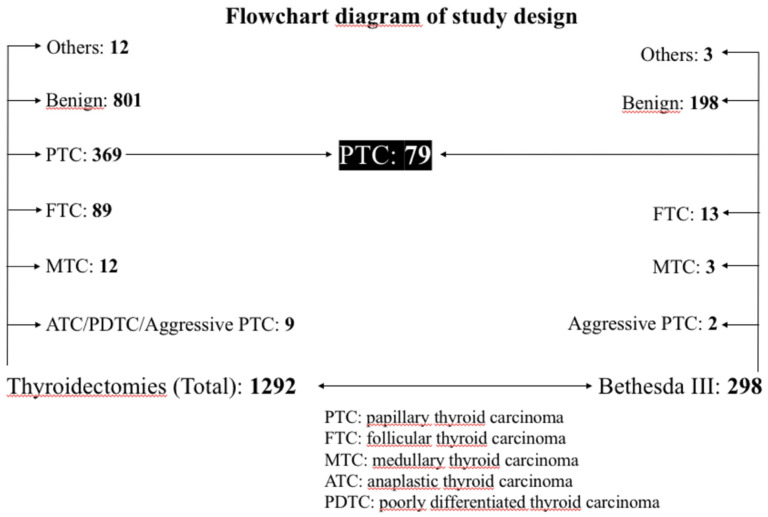
Flowchart diagram of study design. The patients included in the study were recruited, after applying the previously established exclusion criteria, from the total number of PTCs that came from the Bethesda III diagnostic category. Exclusion criteria: FTC—follicular thyroid carcinoma. MTC—medullary thyroid carcinoma. ATC—anaplastic thyroid carcinoma. PDTC—poorly differentiated thyroid carcinoma. Aggressive PTC—Papillary Thyroid Carcinoma other than classical, follicular or incapsulated variant. Others: BIII + indeterminate/suspected/malignant nodule; nodule in Grave’s Disease; neck irradiation.

**Figure 2 medicina-59-01484-f002:**
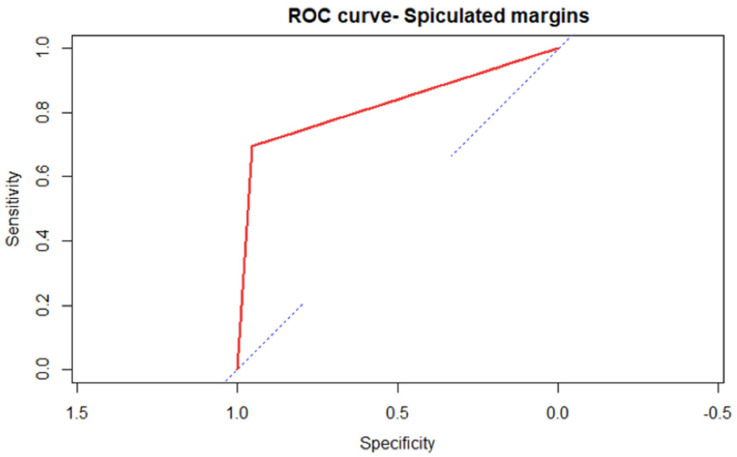
The diagnostic performance of spiculated margins is evidenced by the area under curve, which is visually quite large. Note that the square shape of the ROC curve is a consequence of the analysis performed on a parametric variable.

**Figure 3 medicina-59-01484-f003:**
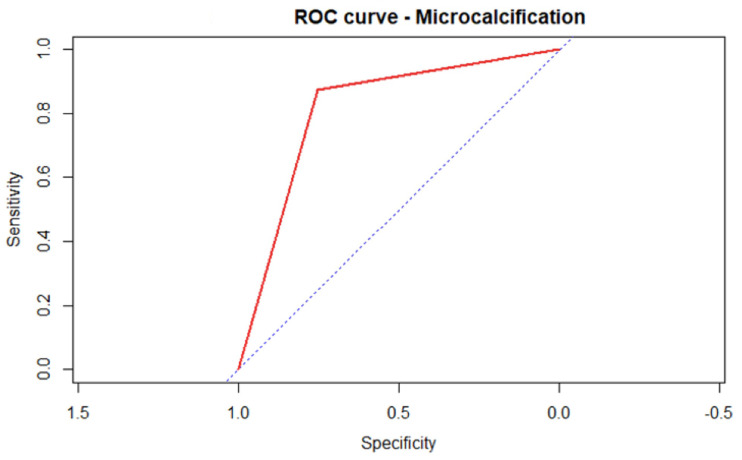
Regarding the variable “microcalcifications” the same indications as the previous one can be applied. The shape of the area under the curve demonstrates, compared with the previous one, a shift in favor of sensitivity.

**Figure 4 medicina-59-01484-f004:**
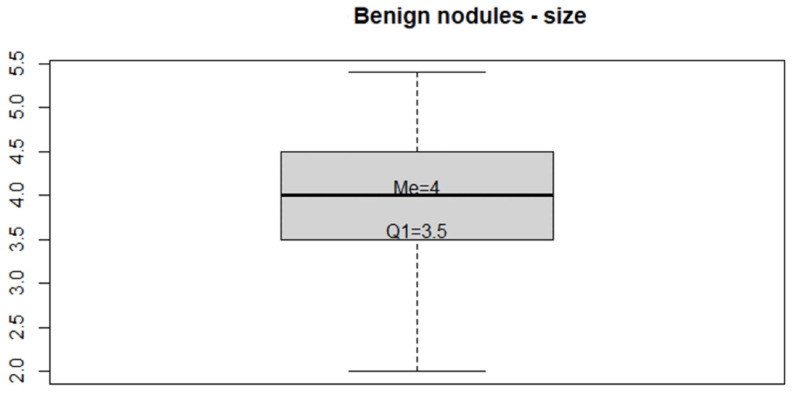
This box-plot represents the size values of benign nodules. The threshold of the first quartile results is 3.5 cm. Being a descriptive statistics tool, this graph is not intended to detect absolute values of statistical significance, but rather to represent the distribution of mean values of this group.

**Table 1 medicina-59-01484-t001:** Univariate analysis.

Variable	Benign	Malignant		OR (CI95%)	*p*-Value
Age	54	55			0.4298489
Sex	Benign	Malignant	Total	OR (CI95%)	*p*-value
m	66 (23%)	22 (7%)	88 (30%)	0.895	0.7743
f	153 (51%)	57 (19%)	210 (70%)		
Total	219 (74%)	79 (26%)	298 (100%)		
Variable	Benign	Malignant		OR (CI95%)	*p*-value
Size	3.85	2.3			<0.00001
Cystic	Benign	Malignant	Total	OR (CI95%)	*p*-value
N	182 (61%)	78 (26.2%)	260 (87.2%)	0.0633 (0.0085–0.4678)	<0.001
Y	37 (12.5%)	1 (0.3%)	38 (12.8%)		
Total	219 (73.5%)	79 (26.5%)	298 (100%)		
Echogenicity	Benign	Malignant	Total	OR (CI95%)	*p*-value
Isohechoic	60 (20.1%)	0 (0%)	60 (20.1%)	13.9	0.000001
Slightly hypohechoic	108 (36.3%)	15 (5%)	123 (41.3%)		
Hypohechoic	51 (17.1%)	64 (21.5%)	115 (38.6%)		
Total	219 (73.5)	79 (26.5%)	298 (100%)		
Intralesional vascularization	Benign	Malignant	Total	OR (CI95%)	*p*-value
N	108 (36.3%)	14 (4.7%)	122 (41%)	4.5 (2.3–9.2)	0.000001
Y	111 (37.2%)	65 (21.8%)	176 (59%)		
Total	219 (73.5%)	79 (26.5%)	298 (100%)		
Spiculated margins	Benign	Malignant	Total	OR (CI95%)	*p*-value
N	209 (70.1%)	24 (8%)	233 (78.1%)	46.7 (20.5–116.9)	0.000001
Y	10 (3.4%)	55 (18.5%)	65 (21.9%)		
Total	219 (73.5%)	79 (26.5%)	298 (100%)		
Microcalcification	Benign	Malignant	Total	OR (CI95%)	*p*-value
N	165 (55.4%)	10 (3.3%)	175 (58.7%)	20.8 (9.8–48.6)	0.000001
Y	54 (18.1%)	69 (23.2%)	123 (41.3%)		
Total	219 (73.5%)	79 (26.5%)	298 (100%)		
Taller than wide	Benign	Malignant	Total	OR (CI95%)	*p*-value
N	218 (73.2%)	54 (18.1%)	272 (91.3%)	99.2 (15.6–4060.8)	0.000001
Y	1 (0.3%)	25 (8.4%)	26 (8.7%)		
Total	219 (73.5%)	79 (26.5%)	298 (100%)		
Extra-thyroid growth	Benign	Malignant	Total	OR (CI95%)	*p*-value
N	219 (73.5%)	54 (18.1)	273 (91.6%)	205.4 (12.3–3427.38)	0.000001
Y	0	25 (8.4%)	25 (8.4%)		
Total	219 (73.5%)	79 (26.5%)	298 (100%)		
Increase	Benign	Malignant	Total	OR (CI95%)	*p*-value
N	66 (22.2%)	14 (4.7%)	80 (26.9%)	2.0 (1.02–4.13)	<0.05
Y	153 (51.3%)	65 (21.8%)	218 (73.1%)		
Total	219 (73.5%)	79 (26.5%)	298 (100%)		

**Table 2 medicina-59-01484-t002:** Reassessment and change strategy value.

Reassessment	Benign	Malignant	Total	OR (CI95%)	*p*-Value
N	11 (3.7%)	0 (0%)	11 (3.7%)	Inf (0.93-Inf)	0.05
Y	208 (69.8%)	79 (26.5%)	287 (96.3%)		
Total	219 (73.5%)	79 (26.5%)	298 (100%)		
Change of strategy	Benign	Malignant	Total	OR (CI95%)	*p*-value
N	197	3	200	218.7 (64.1–1155.2)	<0.00001
Y	22	76	98		
Total	219	79	298		

**Table 3 medicina-59-01484-t003:** Multivariate analysis.

Variable	OR	CI (Inf) 95%	CI (Sup) 95%	*p*-Value
Size	0.193	0.081	0.404	<0.00005 ***
Spiculated margins +	5.694	1.992	10.70	<0.005 **
Microcalcifications +	5.050	1.542	10.88	<0.01 **
Taller than wide +	10.16	1.006	30.09	0.07456
Extra-thyroidal growth +	>6000	>0.001	<20,000	0.98672

AIC: 124.3. ** significant *** highly significativity.

## Data Availability

Data available on request due to restrictions eg privacy or ethical.

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
