# Peer review of "The Role of “Critical” Ultrasound Reassessment in the Decision-Making of Bethesda III Thyroid Nodules"

_medicina, 2023, doi:10.3390/medicina59081484_

Round 1
Reviewer 1 Report
The manuscript titled: The role of "critical" ultrasound reassessment in the decision-making of Bethesda III thyroid nodules, sheds light on topic of thyroid nodules which represents a crucial clinical problem and its evaluation using ultrasound. Indeed, the idea of the manuscript is novel and it is well written using the standard format of Medicina journal. However, the histological data is missing and integration of ultrasound examination and histological findings is also missing. These missing data reduces the quality of the current manuscript.
I recommend the major revision of the current manuscript titled: The role of "critical" ultrasound reassessment in the decision-making of Bethesda III thyroid nodules. The evaluation is based on that overall value of data presented and novelty of the idea in addition to the qulity of manuscript writing. In addition, it is on the scope of this outstanding journal. However, the histological data is missing and integration of ultrasound examination and histological findings is also missing. These missing data reduces the quality of the current manuscript.

Author Response
“The histological data is missing and integration of ultrasound examination and histological findings is also missing. These missing data reduces the quality of the current manuscript”.
Preliminarily, we have added a flowchart clarifying which histotypes were treated in the reporting period, which were included in the study and which are excluded in accordance with the criteria adopted in order to make the sample as homogeneous as possible for better data reliability.
In Table 1, the different ultrasound variables are cross-referenced with the result of the histological examination (benign/malignant). It should be noted that it refers only to the non-aggressive variants of PTC, which is the object of our study.

Reviewer 2 Report
The manuscript highlight roles of ultrasound in the management of Bethesda III thyroid nodule. However, the discussion are not strong to warrant publication at this stage.
There are many statement are incorrect and unclear.
Please refer to attached pdf text with comments for your kind reference.

Please send the manuscript for English editing or native English speaker.
Author Response
Abstract
-Our purpose was to highlight ultrasound criteria suspicious for cancer and evaluate "critical" ultrasound reassessment.
The following sentence was included in the abstract: “Our purpose was to assess which US criteria are most associated with cancer risk, and the value of critical ultrasound (US) reassessment”.
-We also evaluated how and what the ultra-sound re-evaluation modified the strategy.
We added the following sentence:“We also evaluated if the ultrasound reassessment modified the strategy”.
-In the presence of these signs the strategy changed in 76/79 cases of malignancy. Overall, the surgical indication was appropriate in 75% of cases.
We rephrased the sentence as follows: “The presence of these signs readdressed the strategy in 76/79 cases. Then, the indication for surgery was appropriate in 75% of cases”.
Introduction
-Indeed, it has been shown a different impact in neoplastic suspicion in cellular/nuclear atypias compared to architectural ones, since thyroid carcinoma seems to be more frequent in AUS than in FLUS.
We added in the text the following sentence: “In fact, it seems to be clear that AUS would present a higher risk of carcinoma than FLUS.
-As a consequence, BIII represents the main limitation of thyroid cytology, which is nevertheless to be considered the most accurate investigation in the evaluation of the thyroid nodule.
We added in the text the following sentence: “As a result of its heterogeneity, the BIII is one of the major concerns in thyroid cytology, despite the fact that this diagnostic method is to be considered the most accurate investigation in thyroid nodule evaluation”.
-….of the concrete neoplastic risk of any focal lesion on which cytology has not been conclusive.
We added in the text the following sentence : “of cancer suspicion in focal lesions on which cytology has given an inconclusive result.
-….performed at the end of a diagnostic integrated course, can change the choices in favor of surgery on BIII focal lesions,…
We added in the text the following sentence “….repeated after obtaining the result of the cytological examination, would change the choices in favor of surgery in BIII nodules….”
Materials and Methods
-….usually taken assignment by a specific multidisciplinary team established at our institution.
We added in the text the following sentence “…treated by an institutional multidisciplinary team”.
-In particular, surgeons are high-volume as is the institution at which they operate, according to the commonly accepted criteria of the scientific reference societies.
We added in the text the following sentence: “Those enrolled in the present study were operated by high volume surgeons according to the commonly accepted criteria of the scientific reference societies”.
Discussion
-….within which a number of alterations with different meanings coexist.
We added in the text the following sentence “In this category coexists a number of alterations”.
-Nine specific scenarios that may correspond to AUS/FLUS alterations have been reported, which may take the form of a tendency to microfollicular organization, variable representation of Hurtle cells, cellular atypia however ascribable to possible preparation artefacts, up to focal cellular atypia that may even reach papillary carcinoma-like aspects
We added in the text the following sentence “Nine specific groups of changes correspond to AUS/FLUS alterations: microfollicular organization, variable representation of Hurtle cells, cellular atypia attributable to possible preparation artefacts, up to focal cellular atypia that may take on papillary carcinoma-like aspects”.
-…ones analyzed in even more detail.
We added in the text the following sentence “…ones even more analyzed in detail”.
-….very often variously cellular samples do not allow for reliable standardization.
We added in the text the following sentence “…. samples with heterogeneous cellularity do not allow for reliable standardization”
-Especially during the last two decades, molecular biology tests have been developed in the diagnosis of thyroid nodules,….
We added in the text the following sentence “Especially in the last two decades, cytology has been enriched with molecular biology tests, which have improved its initial diagnostic performance”
-….even more so when associated with the previous ones.
We added in the text the following sentence …”has even more value especially if multiple mutations are found in association”.
-The limitations of all these tests in clinical practice lie in the fact that they all have a high throughput if the test is positive, but their negativity is poorly predictive.
We added in the text the following sentence “In clinical practice, molecular biology tests have some limitations. In fact, they all have a high throughput if the test is positive, but their negativity is poorly predictive”.
-…in real-life scenarios, the most common risk scores have been used for the sole purpose of referring detected lesions to FNAB or simple follow-up, depending on the risk score.
We added in the text the following sentence “In current clinical practice, the most common risk scores have been used for the purpose of referring detected lesions to FNAB or follow-up”.
-… Several recent studies have shown that thyroid ultrasonographycanplay a complementary role of great practical value both in predicting the risk of indeterminate thyroid lesions and in ruling them out.
We added in the text the following sentence “Several recent studies have shown that thyroid ultrasonography can also improve the predictive value of lesions found to be indeterminate at cytology”.
-They concluded in favor of helpfulness of ultrasound in differentiating benign from malignant BIII nodules, with a direct proportionality between number of signs present and risk of malignancy.
We added in the text the following sentence “In this study, ultrasound improved the ability to differentiate benign from malignant BIII nodules. The greater the number of ultrasound signs of suspicion, the greater the ability of ultrasound to target malignancy”.
-Moreover, previous studies already looked in the risk upgrade of BIII cases, using ultrasound as criteria, evaluating not only conventional US criteria, but also stiffness.
We added in the text the following sentence “Other studies have assessed the risk of BIII nodules, not only by conventional US criteria, but also stiffness assessed by elastosonography”.
-The identification of highly suspicious US characteristics observed with all techniques increased the risk of malignancy
We added in the text the following sentence “The identification of highly suspicious US characteristics (marked hypoechogenicity, irregular margins, intranodular inhomogeneity, taller-than-wide shape, intranodular microcalcifications, lymph nodes with hilum changes, moderate and severe stiffness showed at elastosonography, moderate or marked perinodular vascularity) observed with all techniques increased the risk of malignancy”.
-Nevertheless, the higher risk of malignancy that nodules with cellular atypia present does not reach statistical significance
We rephrased the sentence as follows:
“Nevertheless, the higher risk of malignancy of nodules with cellular atypia does not reach statistical significance”.
-The first quartile
We changed with “1stquartile”. This word cannot be changed because it is commonly used in statistics.
-In only 25 percent of cases could surgery be classified as an "excess of precaution" or, in any case, a questionable choice relatively to the size of the nodule or associated symptoms.
We added in the text the following sentence: “In conclusion, in our series surgery was an overtreatment in only 25% of cases. This probably occurred because of an overestimation of the symptoms reported by patients, or because of anxiety resulting from knowledge of a pathology that was not well characterized”.
-In the first case, we believe that the current costs do not, at least in the context in which the study was carried out, allow for widespread use of such investigations, even given their reduced rule-out capacity. In the second circumstance, we believe that the continuing uncertainties about the varying significance of the different atypias makes the outcome of any search for statistical correlations uncertain. These may be more difficult to find given the limited number of items in each subclass.
We added the following sentence “Regarding molecular biology investigations, we believe that current costs do not allow for their widespread use. A further limitation is the low negative predictive value. Concerning the BIII subcategories, there are still uncertainties about the different significance of individual atypia. This makes any assessment unreliable”.
-…starting precisely from such a distinction, the refinement of which we hope for in the near future.
We added the following sentence …by applying the distinction between the subcategories of BIII, if this distinction becomes more reliable.
-patient’s pathway
We rephrased the sentence as follows: Patient’s clinical course
-The stepwise analysis approach made it possible to identify a multiple logistic regression model in which the truly independent variables were evaluated, while for the remaining variables, which were significant to the univariate analysis, their possible intercorrelation should be evaluated in further studies on larger case series.
We added the following sentence “The statistical analysis performed identified and evaluated the nondependent variables. We do not exclude that further studies with a larger number of enrolled patients may identify additional significant variables”.
-It is true that molecular biology could further refine this framing, but its cost and availability are not, to date, systematically applicable.
In light of the overall revision process we considered that statement not useful and deleted it.

Reviewer 3 Report
Overall, crude data are interesting.
However, it is mandatory to refine study design description and results presentation.
Remarks
Introduction
- At the beginning of the paragraph, please briefly state the epidemiological scenario: prevalence of thyroid nodules in the overall population by US and prevalence of thyroid cancer among discovered nodules (cite Papini, E.; Guglielmi, R.; Bianchini, A.; Crescenzi, A.; Taccogna, S.; Nardi, F.; Panunzi, C.; Rinaldi, R.; Toscano, V.; Pacella, C.M. Risk of malignancy in nonpalpable thyroid nodules: Predictive value of ultrasound and color-Doppler features. J. Clin. Endocrinol. Metab. 2002, 87, 1941–1946; Parsa, A.A.; Gharib, H. Epidemiology of Thyroid Nodules. In Thyroid Nodules. Contemporary Endocrinology; Gharib, H., Ed. Humana Press: Tortowa, NJ, USA, 2018; pp. 1–11.).
- At the beginning of the paragraph, please also report the main US features increasing oncologic risk of thyroid nodules and the PPV of the most applied US-based systems (refer to Kovatcheva, R.D.; Shinkov, A.D.; Dimitrova, I.D.; Ivanova, R.B.; Vidinov, K.N.; Ivanova, R.S. Evaluation of the Diagnostic Performance of EU-TIRADS in Discriminating Benign from Malignant Thyroid Nodules: A Prospective Study in One Referral Center. Eur. Thyroid J. 2021, 9, 304–312) and accuracy of FNAC evaluation, specifically referring to the prevalence of indeterminate cytology (Cap, J.; Ryska, A.; Rehorkova, P.; Hovorkova, E.; Kerekes, Z.; Pohnetalova, D. Sensitivity and specificity of the fine needle aspiration biopsy of the thyroid: Clinical point of view. Clin. Endocrinol. 1999, 51, 509–515)
- When stating about the role of molecular data, please go more in depth by specifically focusing the role of the most common mutations related to thyroid cancer. Please cite the following papers (Marotta V, Bifulco M, Vitale M. Significance of RAS Mutations in Thyroid Benign Nodules and Non-Medullary Thyroid Cancer. Cancers (Basel). 2021 Jul 27;13(15):3785. doi: 10.3390/cancers13153785. PMID: 34359686; PMCID: PMC8345070; Marotta V, Sapio MR, Guerra A, Vitale M. BRAF mutation in cytology samples as a diagnostic tool for papillary thyroid carcinoma. Expert Opinion on Medical Diagnostics Jul 2011, Vol. 5(4): 277-290).
Methods
- Clearly state the study nature (retrospective, I guess)
- Clearly describe the study design and provide a dedicated figure with a flowchart diagram. For instance, I cannot understand whether authors included a) all patients subjected to thyroidectomy in the study period b) all patients with a Bethesda III cytology subsequently subjected to surgery. This should be cleared to the readers. Finally, report only the data pertinent to the study design.
- Schematically report inclusion and exclusion criteria
- All the data pertinent to the findings (number of enrolled patients, prevalence of the different cytology classes with specific reference to the Bethesda III, histological data) have to be reported within the Results and not in the Methods section.
- Please organize a subparagraph called “Clinical management”, where all pertinent items should be inserted (please shorten the part dedicated to the multidisciplinary approach).
- Please revise statistics. Why Fisher exact test was routinely applied instead of chi-square? Please note that Fisher test is used in substitution of the chi-square only when cells include less than 5 cases.
- Again, statistics: for continuous variables, first should be assessed the distribution (normal or not normal), then should be chosen the test to apply (parametric vs non parametric)
- A relevant aspect is missing. As reported in the aims, authors should clarify whether the analysis has relied on US features as captured at baseline (before cytology) or at the time of US reassessment (namely following the indeterminate results). It should be interesting to compare pre- vs- post cytology US assessment, in order to understand whether a more in depth US evaluation actually impacts on management.
- Please define volume increase. Please discuss that benign nodules, especially if bearing some mutations may have an increase tendency (please cite Puzziello A, Guerra A, Murino A, Izzo G, Carrano M, Angrisani E, Zeppa P, Marotta V, Faggiano A, Vitale M. Benign thyroid nodules with RAS mutation grow faster. Clin Endocrinol (Oxf). 2015 Aug 11. doi: 10.1111/cen.12875. [Epub ahead of print] PubMed PMID: 26260959.)
Results
- How do authors explain that all malignant nodules were PTC? This aspect should be addressed
- Produce a Table with baseline features of enrolled patients
- Please clearly report in the text sensitivity, specificity, PPV, NPV
- Table 1. For each variable state the reference category. I consider to report OR considering as reference the absence of the risk features. Please report some 95% CI intervals that are missing.
- Please try to build a combinatory model of US risk features (a kind of US risk profile) and calculate PPV and NPV, where, considering the clinical scenario, the PPV, therefore the specificity, is the most important parameter, in order to orientate management toward surgery.
Discussion
To address based on previous concerns.
.
Author Response
Introduction
-At the beginning of the paragraph, please briefly state the epidemiological scenario: prevalence of thyroid nodules in the overall population by US and prevalence of thyroid cancer among discovered nodules (cite Papini, E.; Guglielmi, R.; Bianchini, A.; Crescenzi, A.; Taccogna, S.; Nardi, F.; Panunzi, C.; Rinaldi, R.; Toscano, V.; Pacella, C.M. Risk of malignancy in nonpalpable thyroid nodules: Predictive value of ultrasound and color-Doppler features. J. Clin. Endocrinol. Metab. 2002, 87, 1941–1946; Parsa, A.A.; Gharib, H. Epidemiology of Thyroid Nodules. In Thyroid Nodules. Contemporary Endocrinology; Gharib, H., Ed. Humana Press: Tortowa, NJ, USA, 2018; pp. 1–11.).
We have done these changes and added the references suggested
-At the beginning of the paragraph, please also report the main US features increasing oncologic risk of thyroid nodules and the PPV of the most applied US-based systems (refer to Kovatcheva, R.D.; Shinkov, A.D.; Dimitrova, I.D.; Ivanova, R.B.; Vidinov, K.N.; Ivanova, R.S. Evaluation of the Diagnostic Performance of EU-TIRADS in Discriminating Benign from Malignant Thyroid Nodules: A Prospective Study in One Referral Center. Eur. Thyroid J. 2021, 9, 304–312) and accuracy of FNAC evaluation, specifically referring to the prevalence of indeterminate cytology (Cap, J.; Ryska, A.; Rehorkova, P.; Hovorkova, E.; Kerekes, Z.; Pohnetalova, D. Sensitivity and specificity of the fine needle aspiration biopsy of the thyroid: Clinical point of view. Clin. Endocrinol. 1999, 51, 509–515
We have discussed the main aspects of US based on the data from the suggested reference (The main US features ….. to stage 5 (PPV>30%))
-When stating about the role of molecular data, please go more in depth by specifically focusing the role of the most common mutations related to thyroid cancer. Please cite the following papers (Marotta V, Bifulco M, Vitale M. Significance of RAS Mutations in Thyroid Benign Nodules and Non-Medullary Thyroid Cancer. Cancers (Basel). 2021 Jul 27;13(15):3785. doi: 10.3390/cancers13153785. PMID: 34359686; PMCID: PMC8345070; Marotta V, Sapio MR, Guerra A, Vitale M. BRAF mutation in cytology samples as a diagnostic tool for papillary thyroid carcinoma. Expert Opinion on Medical Diagnostics Jul 2011, Vol. 5(4): 277-290).
Done (It has long been known ….. to overcome this problem)
Methods
-Clearly state the study nature
At the beginning of the paragraph we affirmed: “This is a retrospective study … “
-Clearly describe the study design and provide a dedicated figure with a flowchart diagram. For instance, I cannot understand whether authors included a) all patients subjected to thyroidectomy in the study period b) all patients with a Bethesda III cytology subsequently subjected to surgery. This should be cleared to the readers. Finally, report only the data pertinent to the study design.
We provided a flow chart (figure 1) that explains this. Moreover, we affiemed “Therefore, all patients with a Bethesda III cytology subsequently subjected to surgery were recruited for this study”.
-Schematically report inclusion and exclusion criteria
And
-Please organize a subparagraph called “Clinical management”, where all pertinent items should be inserted (please shorten the part dedicated to the multidisciplinary approach).
We have organized a sub-section named as follows. At the end ofthis paragraph , we have added a summary of the inclusion/exclusion criteria. The part dedicated to multidisciplinary approach has been shortened.
-Please revise statistics. Why Fisher exact test was routinely applied instead of chi-square? Please note that Fisher test is used in substitution of the chi-square only when cells include less than 5 cases.
We preferred to use Fisher's exact test for 2 x2 tables because it also allows us to calculate the Odd’s Ratio. This indicator, together with the p-value, allows us to test not only the statistical significance of the relationship, but also its direction. While for the other tables, we applied the chi-square test
- Again, statistics: for continuous variables, first should be assessed the distribution (normal or not normal), then should be chosen the test to apply (parametric vs non parametric)
We apologise but it was a typo because we actually applied Welch's parametric test.
--Please clearly report in the text sensitivity, specificity, PPV, NPV
In particular, the presence of irregular margins appears to be a more specific than sensitive indicator with values respectively: Specificity 0.95 CI95%(0.93-0.98); Sensitivity 0.70 CI95%(0.59-0.80) [Figure 1]. The presence of microcalcifications is a more sensitive than specific indicator with values respectively: Sensitivity 0.87 IC95%(0.80-0.94); Specificity 0.75 IC95%(0.72-0.83) [Figure 3]. This statement is reported in the manuscript, paragraph “results”
-All the data pertinent to the findings (number of enrolled patients, prevalence of the different cytology classes with specific reference to the Bethesda III, histological data) have to be reported within the Results and not in the Methods section.
Given that these data were summarized in the flow chart (Figure 1) we felt that moving them into the results made the argument fragmentary, so we decided not to make this change, as the methods of enrollment seemed quite relevant to the methodology adopted.
- For each variable state the reference category. I consider to report OR considering as reference the absence of the risk features. Please report some 95% CI intervals that are missing.
Done, as requested, in each box of the table 1
-A relevant aspect is missing. As reported in the aims, authors should clarify whether the analysis has relied on US features as captured at baseline (before cytology) or at the time of US reassessment (namely following the indeterminate results). It should be interesting to compare pre- vs- post cytology US assessment, in order to understand whether a more in depth US evaluation actually impacts on management.
We have added the statement “All patients with a BIII cytology Subsequently subjected to surgery (thyroid lobectomy or total thyroidectomy) were included in this study”. Furthermore, from an analysis of Table 2 it can be seen how ultrasound interpretation after critical reassessment may vary.
-Please define volume increase. Please discuss that benign nodules, especially if bearing some mutations may have an increase tendency (please cite Puzziello A, Guerra A, Murino A, Izzo G, Carrano M, Angrisani E, Zeppa P, Marotta V, Faggiano A, Vitale M. Benign thyroid nodules with RAS mutation grow faster. Clin Endocrinol (Oxf). 2015 Aug 11. doi: 10.1111/cen.12875. [Epub ahead of print] PubMed PMID: 26260959.)
We preferred to comment on the volume increase in discussion paragraph. At the end of this analysis, which was based on the 2015 guidelines, we discussed the role of RAS mutation in volume increase and added the suggested reference.
Results
- How do authors explain that all malignant nodules were PTC? This aspect should be addressed.
We think Figure 1 makes it clear that PTC is only one of the variants observed during the reporting period. We limited the study to that variant to employ a homogeneous population and avoid possible bias
-Produce a Table with baseline features of enrolled patients
Table 1 contains these data. Should it be appropriate to add more data, we are ready to review the material in our possession, collected in excel, to derive it.
Round 2
Reviewer 1 Report
I recommend rejection of the manuscript titled: The role of "critical" ultrasound reassessment in the decision-making of Bethesda III thyroid nodules. The evaluation is based on that overall value of data presented and novelty of the idea in addition to the qulity of manuscript writing. In addition, it is on the scope of this outstanding journal. However, the histological data is missing and integration of ultrasound examination and histological findings is also missing. These missing data reduces the quality of the current manuscript.
Author Response
“The histological data is missing and integration of ultrasound examination and histological findings is also missing. These missing data reduces the quality of the current manuscript”.
Preliminarily, we have added a flowchart with its caption in Materials and Methods (Figure 1), clarifying which histotypes were treated in the reporting period, which were included in the study and which are excluded in order to make the sample as homogeneous as possible for better data reliability.
In Materials and Methods, we added the following sentence which is underlined in yellow: “other diagnoses.Therefore, all patients with a Bethesda III cytology subsequently subjected to surgery were recruited for this study.The flowchart in Figure 1 shows how patients were recruited for the study. The 79 operated PTCs from the BIII diagnostic category are part of the total 369 operated PTCs, just as the total 298 BIIIs are part of the 1292 thyroidectomies performed during the reporting period. At the time of ultrasonographic reevaluation, the analysis was based on the ultrasonographic features, namely following the indeterminate results.
In Table 1, the different ultrasound variables are cross-referenced with the result of the histological examination (benign/malignant). It should be noted that it refers only to the non-aggressive variants of PTC, which is the object of our study.
Reviewer 2 Report
The author had made revision as suggested to improve the manuscript
Reviewer 3 Report
I congratulate with the authors for the paper improvement and for addressing my previous indications.
The paper can be accepted once stated the following mandatory suggestions:
- specify in the Methods the timing of first US-asessment and the reassessment;
- specify whether US features reported in the analysis were captured at the time of first US study or at the time of reassessment;
- specify whether there was significat differences in the prevalence of the US-risk features between the first US evaluation and the reassessment. To demonstrate the relevance of a US second-look;
- Table 2 has no meaning. Remove and substitute with the comparison of the prevalence between US risk features between the first US study and the reassessment;
- please clarify the Bethesda edition used by physicians as the Bethesda third Edition of Thyroid cytology has been recently published (Thyroid 2023).
monor revision
Author Response
Introduction
-“At the beginning of the paragraph, please briefly state the epidemiological scenario: prevalence of thyroid nodules in the overall population by US and prevalence of thyroid cancer among discovered nodules (cite Papini, E.; Guglielmi, R.; Bianchini, A.; Crescenzi, A.; Taccogna, S.; Nardi, F.; Panunzi, C.; Rinaldi, R.; Toscano, V.; Pacella, C.M. Risk of malignancy in nonpalpable thyroid nodules: Predictive value of ultrasound and color-Doppler features. J. Clin. Endocrinol. Metab. 2002, 87, 1941–1946; Parsa, A.A.; Gharib, H. Epidemiology of Thyroid Nodules. In Thyroid Nodules. Contemporary Endocrinology; Gharib, H., Ed. Humana Press: Tortowa, NJ, USA, 2018; pp. 1–11.).”
We have done these changes and added the references suggested(Reference 2 and 3).
We added the following sentencehighlighted in yellow in the manuscript:
“ The widespread use of US has raised the prevalence of thyroid nodules to 50% of the general population. Malignancy is found in just over 9% of solitary nodules and 6.3% of multinodular goiters (2) and is influenced by several risk factors (3). Depending on the epidemiological procedure applied, probability that a nodule conceals a carcinoma,”.
-At the beginning of the paragraph, please also report the main US features increasing oncologic risk of thyroid nodules and the PPV of the most applied US-based systems (refer to Kovatcheva, R.D.; Shinkov, A.D.; Dimitrova, I.D.; Ivanova, R.B.; Vidinov, K.N.; Ivanova, R.S. Evaluation of the Diagnostic Performance of EU-TIRADS in Discriminating Benign from Malignant Thyroid Nodules: A Prospective Study in One Referral Center. Eur. Thyroid J. 2021, 9, 304–312) and accuracy of FNAC evaluation, specifically referring to the prevalence of indeterminate cytology (Cap, J.; Ryska, A.; Rehorkova, P.; Hovorkova, E.; Kerekes, Z.; Pohnetalova, D. Sensitivity and specificity of the fine needle aspiration biopsy of the thyroid: Clinical point of view. Clin. Endocrinol. 1999, 51, 509–515
We have discussed the main aspects of US based on the data from the suggested references.
We added the following sentence in “Introduction” and the suggested references (reference 14 and 15):
“The main US features increasing oncologic risk of thyroid nodules of the most applied US-based systemsare microcalcifications, irregular margins, taller-than-wide shape and marked hypoechogenicity, all incorporated in the most widespread US risk score system in Europe is the five-steps EU-TIRADS. This system demonstrated increasing PPV from the early stages (near 0) to stage 5 (PPV > 30%) (14).
It has long been known that cytology is able to discriminate a substantial number of patients with a high probability of malignancy from subjects who have a very low probability (3%) of falling within it (15), however, the indeterminate results suggest the search for solutions to overcome this problem.”
-When stating about the role of molecular data, please go more in depth by specifically focusing the role of the most common mutations related to thyroid cancer. Please cite the following papers (Marotta V, Bifulco M, Vitale M. Significance of RAS Mutations in Thyroid Benign Nodules and Non-Medullary Thyroid Cancer. Cancers (Basel). 2021 Jul 27;13(15):3785. doi: 10.3390/cancers13153785. PMID: 34359686; PMCID: PMC8345070; Marotta V, Sapio MR, Guerra A, Vitale M. BRAF mutation in cytology samples as a diagnostic tool for papillary thyroid carcinoma. Expert Opinion on Medical Diagnostics Jul 2011, Vol. 5(4): 277-290).
Done (It has long been known ….. to overcome this problem)
We added the following sentence “Discussion” and the suggested references (20 and 30):
“This mutation is present in several human cancers, including papillary thyroid carcinoma, and appears to be associated with a poor prognosis, albeit with relatively low specificity (27, 28). Further mutations that can be found in FNAB samples are those related to the telomerase reverse transcriptase (TERT) promoter (29). Mutations of rat sarcoma viral oncogene homolog (RAS) and rearrangement of paired box gene 8-peroxisome/proliferator-activator receptor γ (Pax8/PPARγ)ave also been tested to improve the performance of cytology(30).
Methods
-Clearly state the study nature
At the beginning of the paragraph we affirmed: “This is a retrospective study…”
-Clearly describe the study design and provide a dedicated figure with a flowchart diagram. For instance, I cannot understand whether authors included a) all patients subjected to thyroidectomy in the study period b) all patients with a Bethesda III cytology subsequently subjected to surgery. This should be cleared to the readers. Finally, report only the data pertinent to the study design.
We provided a flow chart (figure 1with caption) that explains this. Moreover, we affirmed “Therefore, all patients with a Bethesda III cytology subsequently subjected to surgery were recruited for this study”.
-Schematically report inclusion and exclusion criteria
And
-Please organize a subparagraph called “Clinical management”, where all pertinent items should be inserted (please shorten the part dedicated to the multidisciplinary approach).
As suggested we organized a sub-section named as follows. At the end of this paragraph, we have added a summary of the inclusion/exclusion criteria. The part dedicated to multidisciplinary approach has been shortened.All added sentences are indicated in yellow in the manuscript:
“In summary, the inclusion/exclusion criteria were as follows:
Inclusion:
-any ‘non aggressive’ PTC recruited in BIII diagnostic category (2018-2022)
-US performed by endocrinologist/endocrine surgeon of multidisciplinary team
Exclusion:
-two or more BIII-IV-V-VI nodules
-US reports not in agreement with ACE/AACE/AME standards
-cytology reports not in agreement with Bethesda system
-Grave’s disease and other benign diseases
-cancers other than PTC
-aggressive variants of PTC
-NIFTP and other tumors of uncertain malignant potential
-PTC<1 cm.”
-Please revise statistics. Why Fisher exact test was routinely applied instead of chi-square? Please note that Fisher test is used in substitution of the chi-square only when cells include less than 5 cases.
We preferred to use Fisher's exact test for 2 x2 tables because it also allows us to calculate the Odd’s Ratio. This indicator, together with the p-value, allows us to test not only the statistical significance of the relationship, but also its direction. While for the other tables, we applied the chi-square test.
- Again, statistics: for continuous variables, first should be assessed the distribution (normal or not normal), then should be chosen the test to apply (parametric vs non parametric)
We apologize but it was a typo because we actually applied Welch's parametric test.
Below is the correction made to the manuscript:
“Welch’s parametrictest for continuous” in “Statistical analysis.”
-Please clearly report in the text sensitivity, specificity, PPV, NPV
In particular, the presence of irregular margins appears to be a more specific than sensitive indicator with values respectively: Specificity 0.95 CI95%(0.93-0.98); Sensitivity 0.70 CI95%(0.59-0.80) [Figure 2]. The presence of microcalcifications is a more sensitive than specific indicator with values respectively: Sensitivity 0.87 IC95%(0.80-0.94); Specificity 0.75 IC95%(0.72-0.83) [Figure 3].This statement is reported in the manuscript, paragraph “results”
-All the data pertinent to the findings (number of enrolled patients, prevalence of the different cytology classes with specific reference to the Bethesda III, histological data) have to be reported within the Results and not in the Methods section.
Given that these data were summarized in the flow chart (Figure 1) we felt that moving them into the results made the argument fragmentary, so we decided not to make this change, as the methods of enrollment seemed quite relevant to the methodology adopted.
- For each variable state the reference category. I consider to report OR considering as reference the absence of the risk features. Please report some 95% CI intervals that are missing.
Done, as requested, in each box of the table 1.
-A relevant aspect is missing. As reported in the aims, authors should clarify whether the analysis has relied on US features as captured at baseline (before cytology) or at the time of US reassessment (namely following the indeterminate results). It should be interesting to compare pre- vs- post cytology US assessment, in order to understand whether a more in depth US evaluation actually impacts on management.
We have added the statement in materials and Methods“all patients with a Bethesda III cytology subsequently subjected to surgery were recruited for this study.”.
Furthermore, from an analysis of Table 2 it can be seen how ultrasound interpretation after critical reassessment may vary.
-Please define volume increase. Please discuss that benign nodules, especially if bearing some mutations may have an increase tendency (please cite Puzziello A, Guerra A, Murino A, Izzo G, Carrano M, Angrisani E, Zeppa P, Marotta V, Faggiano A, Vitale M. Benign thyroid nodules with RAS mutation grow faster. Clin Endocrinol (Oxf). 2015 Aug 11. doi: 10.1111/cen.12875. [Epub ahead of print] PubMed PMID: 26260959.)
We preferred to comment on the volume increase in discussion paragraph. At the end of this analysis, which was based on the 2015 guidelines, we discussed the role of RAS mutation in volume increase and added the suggested reference.
“In our study, we also included, among the variabilities, the volume increase over a time period. This can be defined as suggested in recommendation 23 of 2015 ATA guidelines (20% increase in at least two nodule dimensions with a minimal increase of 2 mm or more than a 50% change in volume) (19).”
We add the reference suggested (49) in Discussion”:”Given the association between volume increase and RAS mutation in benign nodules, it may be useful to perform such genetic evaluation in the future (49).
Results
- How do authors explain that all malignant nodules were PTC? This aspect should be addressed.
We think Figure 2 makes it clear that PTC is only one of the variants observed during the reporting period. We limited the study to that variant to employ a homogeneous population and avoid possible bias
-Produce a Table with baseline features of enrolled patients
Table 1 contains these data. Should it be appropriate to add more data, we are ready to review the material in our possession, collected in excel, to derive it.
Round 3
Reviewer 1 Report
The manuscript titled: The role of "critical" ultrasound reassessment in the decision-making of Bethesda III thyroid nodules, sheds light on topic of thyroid nodules which represents a crucial clinical problem and its evaluation using ultrasound. Indeed, the idea of the manuscript is novel and it is well written using the standard format of Medicina journal. Indeed, the manuscript has been improved.
I recommend the acceptance of the revised version of the manuscript titled: The role of "critical" ultrasound reassessment in the decision-making of Bethesda III thyroid nodules. The evaluation is based on that overall value of data presented and novelty of the idea in addition to the quality of manuscript writing. In addition, it is on the scope of this outstanding journal. Indeed, the manuscript has been improved after the revision to the level to be accepted in this outstanding journal.

Author Response
We would like to thank the reviewer for his interest in our work and his contribution to the improvement of the manuscript with his analysis.
Reviewer 3 Report
No responses to Report 2 have been provided.
Minor issues.
Author Response
I congratulate with the authors for the paper improvement and for addressing my previous indications.
The paper can be accepted once stated the following mandatory suggestions:
- specify in the Methods the timing of first US-assessment and the reassessment.
R: we specified it at the end of the sub-paragraph “thyroid ultrasound” of “Materials and methods” and the sentence isunderlined in yellow: The first US assessment was performed, therefore, at the time of patient recruitment by the multidisciplinary team. US reassessment was performed after obtaining the results of cytology.US features reported in the analysis were captured at the time of reassessment.
- specify whether US features reported in the analysis were captured at the time of first US study or at the time of reassessment.
R: we clarified next that the analysis of ultrasound signs was performed on the reassessment data
- specify whether there was significant differences in the prevalence of the US-risk features between the first US evaluation and the reassessment. To demonstrate the relevance of a US second-look.
R: we did not find it appropriate to make such a comparison on the data available so we included this in the study's weaknesses and explained why in “Discussion”: “Given the heterogeneous origin of the US at the first evaluation, we did not compare individual variables before and after reevaluation. Moreover, for some of them (e.g., "taller than wide" appearance, "extra-thyroid growth"), the absence of items in the benign category would have invalidated the results”.
- Table 2 has no meaning. Remove and substitute with the comparison of the prevalence between US risk features between the first US study and the reassessment.
R: due to the same reason explained in the previous point, it was not possible to make a table comparing individual items. Instead, we made a comparison on the aggregate data, assessing how often and in which cases we made a reassessment, and how many times and in what context this resulted in a change of strategy.
- please clarify the Bethesda edition used by physicians as the Bethesda third Edition of Thyroid cytology has been recently published (Thyroid 2023).
R: Since the study started on data available from 2018, thyroid cytopathology was uniformly evaluated in accordance with first Bethesda System, as clarified in the manuscript.